# Impact of the COVID-19 nonpharmaceutical interventions on influenza and other respiratory viral infections in New Zealand

Q. Sue Huang [1✉], Tim Wood[1], Lauren Jelley[1], Tineke Jennings[2], Sarah Jefferies[1], Karen Daniells[1], Annette Nesdale[2], Tony Dowell[3], Nikki Turner[4], Priscilla Campbell-Stokes[2], Michelle Balm[5], Hazel C. Dobinson [5], Cameron C. Grant[4], Shelley James[5], Nayyereh Aminisani[1], Jacqui Ralston[1], Wendy Gunn[1], Judy Bocacao[1], Jessica Danielewicz[1], Tessa Moncrieff[1], Andrea McNeill[1], Liza Lopez[1], Ben Waite[1], Tomasz Kiedrzynski[6], Hannah Schrader[1], Rebekah Gray [1], Kayla Cook[1], Danielle Currin[1], Chaune Engelbrecht[2], Whitney Tapurau[2], Leigh Emmerton[2], Maxine Martin[2], Michael G. Baker[3], Susan Taylor[7], Adrian Trenholme[7], Conroy Wong[7], Shirley Lawrence[7], Colin McArthur[8], Alicia Stanley[8], Sally Roberts[8], Fahimeh Rahnama[8], Jenny Bennett[9], Chris Mansell[9], Meik Dilcher[10], Anja Werno[10], Jennifer Grant[11], Antje van der Linden[11], Ben Youngblood [12], Paul G. Thomas [12] & NPIsImpactOnFlu Consortium* & Richard J. Webby[12]

Stringent nonpharmaceutical interventions (NPIs) such as lockdowns and border closures are not currently recommended for pandemic influenza control. New Zealand used these NPIs to eliminate coronavirus disease 2019 during its first wave. Using multiple surveillance systems, we observed a parallel and unprecedented reduction of influenza and other respiratory viral infections in 2020. This finding supports the use of these NPIs for controlling pandemic influenza and other severe respiratory viral threats.

[1] Institute of Environmental Science and Research, Wellington, New Zealand. [2] Regional Public Health, Hutt Valley District Health Board, Wellington, New Zealand. [3] University of Otago, School of Medicine in Wellington, Wellington, New Zealand. [4] University of Auckland, Auckland, New Zealand. [5] Capital Coast District Health Board, Wellington, New Zealand. [6] Ministry of Health, Wellington, New Zealand. [7] Counties Manukau District Health Board, Auckland, New Zealand. [8] Auckland District Health Board, Auckland, New Zealand. [9] Waikato District Health Board, Hamilton, New Zealand. [10] Canterbury District Health Board, Christchurch, New Zealand. [11] Southern District Health Board, Dunedin, New Zealand. [12] WHO Collaborating Centre, St Jude Children's Research Hospital, Memphis, TN, USA. *A list of authors and their affiliations appears at the end of the paper. ✉email: sue.huang@esr.cri.nz

The coronavirus disease 2019 (COVID-19), declared a pandemic by the World Health Organisation (WHO) on 11 March 2020, was first identified in a person in New Zealand (NZ) on 28 February 2020. From 2 February 2020, NZ introduced progressive border restrictions and a four-level alert system aiming to eliminate COVID-19[1]. Soon after the emergence of community transmission of COVID-19, the stringent nonpharmaceutical interventions (NPIs) of Level-4 (nationwide lockdown) were applied, lasting from 25 March to 27 April 2020. These included (1) blocking importation of the virus (border closure to non-New Zealanders and 14-day quarantine for returning travellers); (2) stamping out transmission within NZ (widespread testing, isolating cases, contact tracing and quarantine of exposed persons); (3) physical distancing measures (stay-at-home orders, cancelling all gatherings, closing schools, non-essential businesses and all public venues and restricting domestic travel); (4) individual infection prevention and control measures (promoting hand hygiene and cough etiquette, staying home with mild respiratory symptoms and mask wearing if unwell); and (5) communicating risk to the public and various stakeholders. The implementation of these NPIs combined with public compliance effectively eliminated community transmission of COVID-19 during the first wave (12 February to 13 May 2020), achieving 101 consecutive days without detection of community COVID-19 cases[2,3]. Since this implementation, NZ has continued to apply NPIs in various forms up until submission of this report[1].

The effectiveness of NPIs in reducing viral transmission depends on transmission characteristics of the virus[4]. If a substantial proportion of transmission occurs before the onset of symptoms (i.e. pre-symptomatic shedding) or during asymptomatic infection, the population impact of health screening and case-patient isolation will be diminished[5]. Influenza virus has a short serial interval (the mean interval between illness onset in two successive patients in a chain of transmission) of 2–4 days. Viral excretion peaks early in the illness (i.e. during the first 1–3 days of illness). These features of influenza infection mean there is limited time to effectively implement isolation and quarantine measures. Additionally, substantial asymptomatic infection[6] creates difficulties in finding cases to initiate nonpharmaceutical measures. These characteristics have led to the assumption that these NPIs would not be effective in controlling influenza virus[7]. However, robust field data are lacking. New Zealand's use of stringent NPIs created a natural experiment enabling an understanding of the impact of these NPIs on illnesses caused by influenza and other respiratory viruses. This type of knowledge is valuable for informing pandemic influenza preparedness and seasonal influenza planning for the northern hemisphere's upcoming winter in the context of the ongoing COVID-19 pandemic.

Here we describe the complete absence of the usual winter influenza virus epidemic and a remarkable reduction of other respiratory viral infections in NZ during and after the implementation of these stringent NPIs in 2020.

## Results

Influenza activity in NZ during the winter of 2020 was very low as confirmed by multiple national surveillance systems (Fig. 1).

From May to September 2020, hospital-based severe acute respiratory illness (SARI) surveillance (catchment population of 1 million people) showed very low SARI incidence rates, all below the seasonal threshold defined by the reference period of 2015–2019 (Fig. 1a). No influenza-associated SARI was identified (Fig. 1b).

The national sentinel general practice (GP)-based surveillance usually covers ~10% of the NZ population and captures patients with influenza-like illness (ILI) attending medical consultations. However, this patient flow was altered in 2020 as many patients

with ILI were channelled to COVID-specific testing centres where patients were predominantly only tested for severe acute respiratory syndrome coronavirus 2 (SARS-CoV-2). Additionally, the number of participating practices were 18–57% lower than the usual participation rate over the surveillance period. The ILI incidence rates were below the seasonal threshold compared to the reference period (Fig. 1c). No influenza-associated ILI were detected (Fig. 1d). Independently, the low ILI incidence rates were also observed in the HealthStat GP-based ILI surveillance system (Supplementary Fig. 1).

SHIVERS-II&III (the second and third iterations of the Southern Hemisphere Influenza and Vaccine Effectiveness Research and Surveillance programme) are two community-based cohorts that follow ~1400 adults aged 20–69 years and ~80 infants in the Wellington region, respectively. Active surveillance for both cohorts in 2020 included swabbing of participants meeting the case definition for ILI and/or acute respiratory illness (ARI). The ILI incidence rate in 2020 was lower than the previous years of 2019 and 2018; however, ARI incidence was high (Fig. 1e). No influenza-associated ILI or ARI were identified (Fig. 1f).

The national International Classification of Diseases (ICD)-coded (ICD-10AM-VI code J9-J11) influenza hospitalisations for all NZ public hospitals showed a significant decline (Fig. 1g). From 1 January to 31 July 2020, a total of 291 influenza hospitalisations were coded: pre-lockdown 238 (81.8%), lockdown 33 (11.3%), and post-lockdown 15 (5.2%). The Cochran–Armitage test showed a significant downward trend ($p < 0.001$) in influenza hospitalisations.

The laboratory-based surveillance system includes testing samples ordered by clinicians during routine clinical diagnostic processes for hospital inpatients and outpatients (serving ~70% of the NZ population). Additionally, this system also includes testing samples from all influenza surveillance systems (Fig. 1h). During the COVID-19 laboratory response, some laboratories may have prioritised testing for SARS-CoV-2 over influenza and other respiratory viruses. From 1 January to 27 September 2020, there were 500 influenza virus detections: pre-lockdown 474 (94.8%), lockdown 20 (4.0%), and post-lockdown 6 (1.2%). The Cochran–Armitage test showed a significant downward temporal trend ($p < 0.001$) in influenza virus detections.

Table 1 shows the number of respiratory viruses detected and the proportional reduction for each virus in 2020 (versus the reference period of 2015–2019) before, during and after the lockdown. Dramatic reductions were observed for influenza virus compared with the reference period: 67.7% reduction during and 99.9% after the lockdown. Marked reductions were also evident for other respiratory viruses during post-lockdown compared with the reference period (for temporal distribution, see Supplementary Fig. 2): respiratory syncytial virus (RSV; 98.0% reduction), human metapneumovirus (hMPV; 92.2%), enterovirus (82.2%), adenovirus (81.4%), parainfluenza virus types 1–3 (PIV1–3; 80.1%), and rhinovirus (74.6%). During post-lockdown when the restrictions were eased to Level-1, we observed a significant increase in the proportion of rhinovirus compared to the median rate for this period from the preceding period: 33% (175/529) from 8 June to 11 August 2020 vs. 4.8% (10/209) from 13 May to 7 June 2020 ($p < 0.0001$). The rhinovirus-associated incidence rates in 2020 among SHIVERS-II&III and SARI surveillance also increased after the ease of restrictions (Supplementary Fig. 3).

## Discussion

New Zealand, a southern hemisphere country with a temperate climate, has a well-established influenza circulation pattern with peak incidences in the winter months[8]. Multiple surveillance systems showed that there was no annual laboratory-confirmed

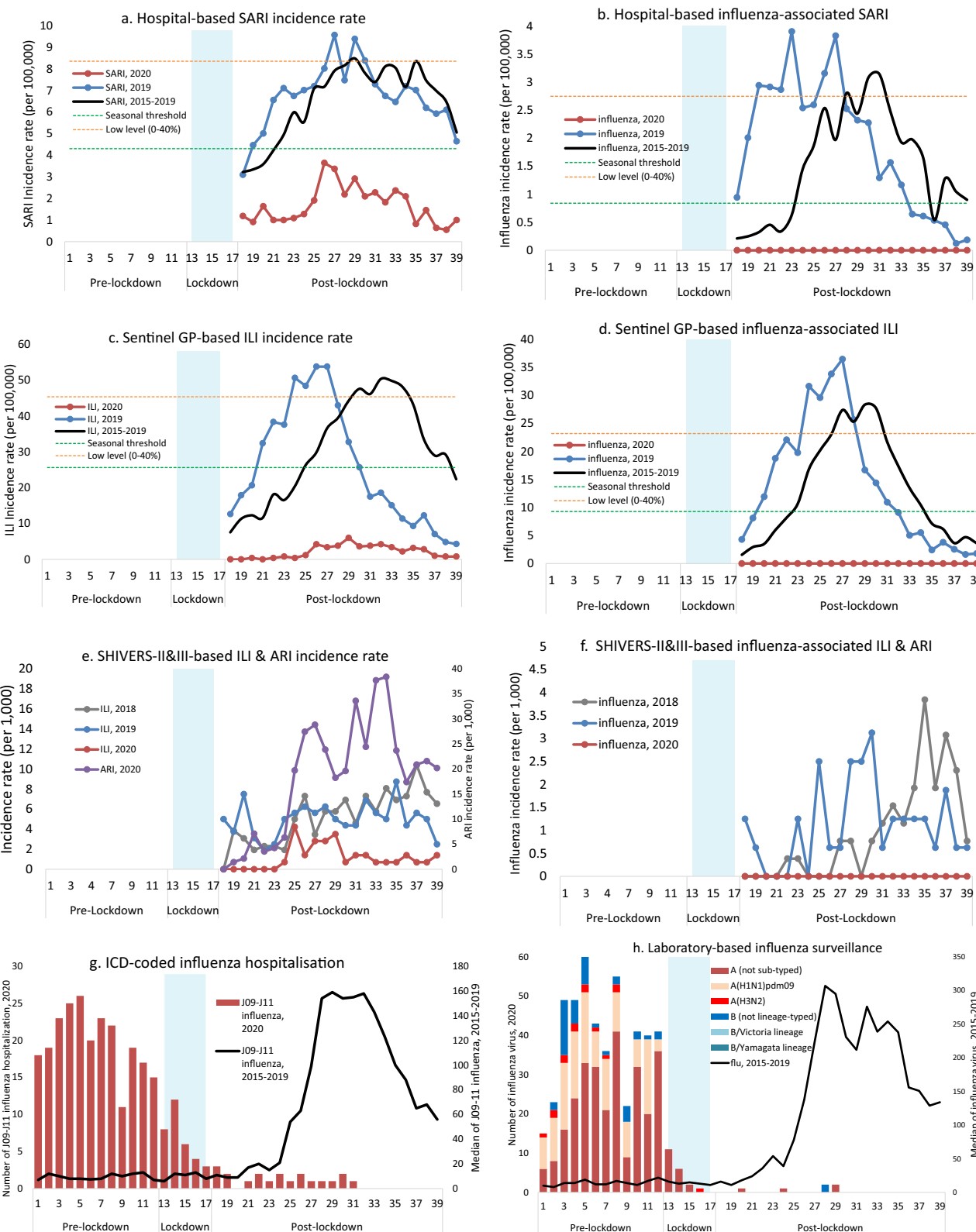

influenza outbreak or epidemic detected during the 2020 winter season. Remarkably, influenza virus circulation was almost non-existent during the 2020 winter, a 99.9% reduction compared with previous years. We postulate that NZ's use of stringent NPIs (lockdowns and border controls) have markedly changed human behaviour[3], resulting in substantial reductions in contacts between influenza-infected individuals and influenza-susceptible

individuals. The nationwide lockdown occurred during late autumn before the usual influenza season. This timing may also be important as the resulting small number of influenza-infected individuals did not appear sufficient to trigger a sustained influenza epidemic in the oncoming winter in a setting of strict border control, personal hygiene promotion and ongoing forms of social distancing that remained in place after the lockdown[1].

**Fig. 1 Temporal distribution of acute respiratory illness and associated influenza infections in 2020 compared with previous years. a, b** Hospital-based SARI incidence rate and influenza-associated SARI. **c, d** Sentinel GP-based ILI incidence rate and influenza-associated ILI. **e, f** SHIVERS-II&III-based ILI and ARI incidence rate and influenza-associated ILI and ARI. **g** ICD-coded influenza hospitalisation. **h** Laboratory-based influenza surveillance. SARI severe acute respiratory illness, GP general practice, ILI influenza-like illness, ARI acute respiratory illness, SHIVERS-II&III the second and third iterations of the Southern Hemisphere Influenza and Vaccine Effectiveness Research and Surveillance programme, ICD International Classification of Diseases, flu refers to influenza. The calculation for epidemic threshold and low influenza activity is described in "Methods". A patient with cough and history of fever (subjective fever or measured temperature ≥38 °C) and onset within the past 10 days meets the SARI case definition if hospitalised or meets the ILI case definition if consulting a GP or participating in the SHIVERS-II&III study. The ARI case definition among SHIVERS-II&III participants refers to an acute respiratory illness with fever or feverishness and/or one of the following symptoms (cough, running nose, wheezing, sore throat, shortness of breath, loss of sense of smell/taste) with onset in the past 10 days.

The WHO's pandemic influenza intervention guidance does not recommend stringent NPIs when pandemic influenza reaches sustained transmission in the general population because these NPIs have been considered ineffective and impractical[5]. However, the knowledge base used in developing WHO guidance for influenza pandemic prevention is limited and consists primarily of historical observations and modelling studies. NZ data, presented here, is consistent with what reported from other southern hemisphere countries[9,10] in Australia, Chile and South Africa, as well as reported from Hong Kong during the 2003 SARS epidemic[11] and the COVID-19 pandemic[12]. Therefore, we suggest that it is important to re-evaluate the role of stringent NPIs such as lockdowns and border closures in mitigating or even eliminating severe pandemic influenza. Although such measures are associated with significant negative impacts on society, their potential beneficial effects on delaying, containing or averting transmission and saving lives should be assessed. New knowledge from this assessment may inform better preparedness for future influenza pandemics and other severe respiratory viral threats. Additionally, it would be a worthwhile endeavour to conduct detailed analysis to identify which components of NPIs were most effective for preventing seasonal influenza and other respiratory virus infection and transmission. Careful investigation of NPIs may identify new and sustainable interventions that can minimise and prevent seasonal and epidemic respiratory viral illnesses in the future.

Other potential contributing factors for the reduction in influenza virus detections include influenza vaccination, climatic changes and viral–viral interactions. The NZ National Immunisation Register recorded ~22% influenza vaccine coverage in 2020 (35% more influenza vaccinations recorded during April–June in 2020 compared with 2019, personal communication). Cold temperature promotes the ordering of lipids on the viral membrane, which increases the stability of the influenza virus particle[13]. Winter 2020 was NZ's warmest winter on record. The nationwide average temperature was 9.6 °C, 1.1 °C above the 1981–2010 average[14]. The warmer winter may reduce virus stability, contributing to lower influenza circulation. A number of viral–viral interactions may also be influencing the incidence of respiratory virus infections. Interferon-stimulated immunity caused by one virus infection can provide non-specific interference making it difficult for additional viruses to become established in a population. Increased levels of influenza A virus circulation have been shown to limit rhinovirus prevalence, potentially through an interferon-mediated mechanism[15]. Others have suggested that the opposite may also be true where rhinovirus circulation can limit influenza virus activity as was suggested in Sweden and France during the 2009 H1N1 pandemic that the annual autumn rhinovirus epidemic interrupted and delayed community transmission of the emerging influenza virus[16,17]. The increase in rhinovirus detection after the lockdown that we noted here may have contributed to an absence of influenza virus circulation.

Stringent NPIs may contribute to the significant reduction of all other non-influenza respiratory viral infections, including RSV, hMPV, PIV1–3, adenovirus, enterovirus and rhinovirus. Unlike the report from the Sydney Children's Hospital Network[18] where an increase in RSV detections occurred at the tail end of the winter season, NZ did not see any increase in RSV detections during the whole 2020 winter season. When the NPIs were relaxed after lockdown, the incidence of rhinovirus increased rapidly, a trend not seen with these other viruses. The mechanism behind this finding is unclear. Rhinovirus infections, responsible for more than one-half of cold-like illnesses, are frequently transmitted within households from children to other family members[19]. Additionally, rhinoviruses are non-enveloped viruses so might be inherently less susceptible to inactivation by soap-and-water handwashing[18]. Furthermore, the quality of children's handwashing is likely to be poor. These factors may have contributed to rhinovirus infection being less affected by the COVID-19 control measures.

In the incoming autumn and winter of 2020 and 2021, many northern hemisphere temperate countries will have continuing COVID-19 circulation overlapping with the influenza season, resulting in increased burden on already stretched health systems. NZ's experience strongly suggests that NPIs can greatly reduce the intensity of seasonal influenza and other respiratory viral infections. Continuation or strengthening of NPIs may, therefore, have positive impacts far beyond COVID-19 control. Even without these interventions, the severity of the 2020–2021 northern hemisphere influenza season remains uncertain. Both international and domestic air travel has been suggested as important drivers of influenza introduction and subsequent spread[20]. It is possible that fewer seeding events from NZ and other southern hemisphere countries, from both reduced influenza activity and reduced air travel, may result in low influenza activity in these northern hemisphere countries during their incoming winter.

Our study has several limitations. First, this is an observational study. Multiple simultaneous measures were applied depending on alert levels, making it difficult to understand the relative contribution of each of these measures. Second, during the COVID-19 laboratory response, some laboratories prioritised testing for COVID-19 and reduced testing for influenza and other respiratory viruses. Additionally, those samples ordered by clinicians for hospital inpatients and outpatients during normal clinical practices were based on clinician's judgement, rather than a systematic sampling approach. This may result in selection bias. Third, the government set up a number of community-based testing centres around the country to provide access to safe and free sampling for COVID-19. The usual flow and processes established for sentinel GP-based ILI surveillance may have been interrupted as many ILI patients would visit these centres instead of sentinel GP clinics. Additionally, national sentinel GP-based ILI surveillance requires swabbing from an ILI patient. This may contribute to the lower GP participation for this surveillance during the COVID-19 pandemic. These factors probably resulted in lower consultation and reporting and sample collection for sentinel ILI surveillance in 2020. However, the SARI surveillance system and SHIVERS-II&III cohorts

**Table 1 The number of influenza and other respiratory viruses detected and their reduction in 2020 before, during and after the lockdown compared with the reference period of 2015-2019.**

| Virus | Pre-lockdown | | | | Lockdown | Post-lockdown | | | | | Subtotal |
|---|---|---|---|---|---|---|---|---|---|---|---|
| | | | | | Level 4 | Level 3 | Level 2 | Level 1 | Level 2&3 | Level 2&1 | |
| Month | Jan | Feb | 1-24 Mar | Subtotal | 25 Mar-27 Apr | 28 Apr-12 May | 13 May-7 Jun | 8 Jun-11 Aug | 12 Aug-29 Aug | 31 Aug-27 Sept | |
| No. of weeks | 1-5 | 6-9 | 10-12 | | 13-17 | 18-19 | 20-23 | 24-32 | 33-35 | 36-39 | |
| **Influenza** | | | | | | | | | | | |
| No. (2020) | 196 | 156 | 122 | 474 | 20 | 0 | 1 | 5 | 0 | 0 | 6 |
| Median (2015-2019) | 84 | 55 | 62 | 206 | 62 | 27 | 126 | 2618 | 731 | 544 | 5062 |
| IQR (2015-2019) | 39-85 | 30-94 | 21-66 | 90-240 | 51-139 | 20-79 | 89-318 | 1120-3165 | 549-996 | 513-935 | 3131-5097 |
| %Reduction[a] | -133.3 | -183.6 | -96.8 | -130.1 | 67.7 | 100.0 | 99.2 | 99.8 | 100.0 | 100.0 | 99.9 |
| **RSV** | | | | | | | | | | | |
| No. (2020) | 9 | 16 | 9 | 34 | 16 | 3 | 8 | 15 | 5 | 3 | 34 |
| Median (2015-2019) | 14 | 15 | 15 | 44 | 85 | 46 | 162 | 1176 | 270 | 159 | 1743 |
| IQR (2015-2019) | 14-15 | 11-16 | 15-20 | 40-45 | 43-93 | 40-57 | 159-167 | 1056-1219 | 253-361 | 146-287 | 1591-2103 |
| %Reduction[a] | 35.7 | -6.7 | 40.0 | 22.7 | 81.2 | 93.5 | 95.1 | 98.7 | 98.1 | 98.1 | 98.0 |
| **Rhinovirus** | | | | | | | | | | | |
| No. (2020) | 103 | 177 | 238 | 518 | 26 | 2 | 10 | 175 | 95 | 44 | 326 |
| Median (2015-2019) | 72 | 79 | 90 | 232 | 144 | 79 | 209 | 529 | 175 | 305 | 1282 |
| IQR (2015-2019) | 67-85 | 76-87 | 77-111 | 220-303 | 135-193 | 62-94 | 206-227 | 498-538 | 172-216 | 244-336 | 1277-1384 |
| %Reduction[a] | -43.1 | -124.1 | -164.4 | -123.3 | 81.9 | 97.5 | 95.2 | 66.9 | 45.7 | 85.6 | 74.6 |
| **Enterovirus** | | | | | | | | | | | |
| No. (2020) | 34 | 30 | 23 | 87 | 3 | 0 | 5 | 33 | 9 | 14 | 61 |
| Median (2015-2019) | 40.5 | 34 | 34 | 107.5 | 69 | 26.5 | 44 | 110.5 | 53.5 | 84 | 343 |
| IQR (2015-2019) | 34-48 | 29-45 | 25-54 | 102-133 | 58-81 | 20-35 | 42-51 | 88-140 | 41-56 | 67-107 | 262-383 |
| %Reduction[a] | 16.0 | 11.8 | 32.4 | 19.1 | 95.7 | 100.0 | 88.6 | 70.1 | 83.2 | 83.3 | 82.2 |
| **Adenovirus** | | | | | | | | | | | |
| No. (2020) | 54 | 54 | 37 | 145 | 9 | 3 | 12 | 39 | 25 | 24 | 103 |
| Median (2015-2019) | 83 | 59.5 | 66.5 | 207 | 75 | 46.5 | 79 | 227.5 | 84.5 | 129 | 554 |
| IQR (2015-2019) | 45-151 | 53-77 | 54-76 | 156-299 | 73-100 | 41-55 | 70-105 | 207-290 | 75-131 | 89-237 | 519-780 |
| %Reduction[a] | 34.9 | 9.2 | 44.4 | 30.0 | 88.0 | 93.5 | 84.8 | 82.9 | 70.4 | 81.4 | 81.4 |
| **hMPV** | | | | | | | | | | | |
| No. (2020) | 11 | 26 | 45 | 82 | 10 | 3 | 10 | 17 | 7 | 3 | 40 |
| Median (2015-2019) | 7 | 4 | 6 | 16 | 10 | 5 | 24 | 187 | 103 | 159 | 513 |
| IQR (2015-2019) | 7-10 | 2-5 | 4-10 | 12-21 | 5-13 | 5-8 | 20-26 | 161-207 | 95-109 | 147-178 | 458-538 |
| %Reduction[a] | -57.1 | -550.0 | -650.0 | -412.5 | 0.0 | 40.0 | 58.3 | 90.9 | 93.2 | 98.1 | 92.2 |
| **PIV** | | | | | | | | | | | |
| No. (2020) | 25 | 25 | 22 | 72 | 32 | 9 | 24 | 41 | 18 | 9 | 101 |
| Median (2015-2019) | 27 | 9 | 7 | 43 | 28 | 17 | 31 | 230 | 82 | 148 | 508 |
| IQR (2015-2019) | 24-28 | 8-12 | 6-8 | 38-45 | 21-37 | 16-19 | 31-55 | 200-317 | 66-148 | 136-231 | 398-803 |
| %Reduction[a] | 7.4 | -177.8 | -214.3 | -67.4 | -14.3 | 47.1 | 22.6 | 82.2 | 78.0 | 93.9 | 80.1 |

New Zealand COVID-19 alert levels and dates and related public health measures can be accessed at: https://covid19.govt.nz/assets/resources/tables/COVID-19-alert-levels-summary.pdf.
IQR interquartile range, RSV respiratory syncytial virus, hMPV human metapneumovirus, PIV parainfluenza virus types 1-3.
a%Reduction = 1 − [no. of virus (2020)/median no. of virus (2015-2019)]%.

operated as usual and showed the same apparent elimination of influenza virus circulation.

In conclusion, this observational study reported an unprecedented reduction in influenza and other important respiratory viral infections and the complete absence of an annual winter influenza epidemic, most likely due to the use of stringent NPIs (border restrictions, isolation and quarantine, social distancing and human behaviour changes). These data can inform future pandemic influenza preparedness and seasonal influenza planning for the northern hemisphere's upcoming winter.

## Methods

**Ethical approval.** Ethical approval was obtained for the SHIVERS (including SARI and ILI surveillance), SHIVERS-II and III cohort studies from the NZ Northern A Health and Disability Ethics Committee (NTX/11/11/102). The ICD-coded influenza hospitalisation data and laboratory-based respiratory virus surveillance data are part of public health surveillance in NZ. It is conducted in accordance with the Public Health Act and thus ethics committee approval was not needed for collection or use of these data.

**Hospital-based SARI surveillance.** The population-based hospital surveillance for SARI among residents (ca ~1 million) of Central (Auckland District Health Board) and South (Counties Manukau District Health Board) Auckland was established in 2012[21]. Research nurses reviewed daily records of all overnight acutely admitted inpatients to identify any inpatient with a suspected ARI. They enrolled those patients with cough and history of fever (subjective fever or measured temperature ≥38 °C) and onset within the past 10 days, defined by the World Health Organisation as SARI. A respiratory specimen (nasopharyngeal or nasal or throat swab) was collected and tested simultaneously for influenza and other respiratory viruses by real-time reverse transcription polymerase chain reaction (PCR) techniques[22]: influenza virus, RSV, rhinovirus, PIV1–3, enterovirus, adenovirus, hMPV.

**Sentinel GP-based ILI surveillance.** The population-based surveillance for ILI among persons enrolled in sentinel GPs (~90) who seek medical consultations has been in operation since 1990[8], usually covering ~10% of the NZ population. The participating general practitioners and practice nurses assessed all consultation-seeking patients. If a patient met the ILI case definition: "an ARI with a history of fever or measured fever of ≥38 °C, and cough, and onset within the past 10 days, and requiring consultation in that GP", a respiratory specimen (nasopharyngeal or nasal or throat swab) was collected to test for influenza and other respiratory viruses[21]. In 2020, sentinel GP-based ILI surveillance was not operated in the usual way due to the COVID-19 response. Instead of visiting sentinel GPs for medical consultations, many ILI patients would visit one of the community-based COVID-19 testing centres. Additionally, national sentinel GP-based ILI surveillance requires swabbing from an ILI patient. This may contribute to less GP participation (18–57% of the usual participation rate over the winter period in 2020) in the COVID-19 pandemic situation. These factors would contribute to lower consultation for, reporting and detection of influenza and other respiratory viruses compared with previous years.

**SHIVERS-II and WellKiwis cohort ILI surveillance.** SHIVERS-II is a prospective adult cohort study in Wellington, NZ. The cohort study has been in operation since 2018 enrolling individuals aged 20–69 years, randomly selected from those healthy individuals listed in the GP's primary care management system. In 2020, SHIVERS-II study staff followed these participants (~1400) and monitored their ILIs and ARIs.

WellKiwis (i.e. SHIVERS-III) is a prospective Wellington infant cohort aiming to recruit 600 infant–mother pairs from Oct 2019 to Sept 2022 (200 pairs a year) and follow them until 2026. In 2020, WellKiwis study staff followed up ~80 infants and monitored their ILIs and ARIs.

During May–September 2020, SHIVERS-II and WellKiwis study staff sent weekly surveys to participants regarding their respiratory illness. Due to COVID-19, the ARI case definition in 2020 has changed to: "ARI with fever or feverishness and/or one of the following symptoms (cough, running nose, wheezing, sore throat, shortness of breath, loss of sense of smell/taste) with onset in the past 10 days". The case definition for ILI during 2018–2020 was the same: ARI with cough and fever/measured fever of ≥38 °C and onset within the past 10 days. For those participants who met the case definition for ILI and ARI, research nurses visited the participant and took a nasopharyngeal or nasal or throat swab to test for influenza and other respiratory viruses and SARS-CoV-2[23].

**ICD-coded influenza hospitalisations.** Hospitalisation data for ICD-coded influenza hospitalisations (ICD-10AM-VI codes (J09–J11) were extracted from the NZ Ministry of Health's National Minimum Dataset by discharge date. In this data set, patients who spent <1 day in a hospital are excluded. Influenza-related hospitalisations are conservatively taken to include only those cases where influenza was the principal diagnosis. Repeat hospital admissions were included because infection with a different influenza A sub-type or influenza B virus is possible.

**Laboratory-based surveillance.** The laboratory-based surveillance for influenza and common respiratory viruses is carried out all year around by the NZ virus laboratory network consisting of the WHO National Influenza Centre at the Institute of Environmental Science and Research and six hospital laboratories in Auckland (2), Waikato, Wellington, Christchurch and Dunedin. This laboratory network tests specimens ordered by clinicians for hospital inpatients and outpatients during normal clinical practice (serving ~70% of the NZ population). Sample collection is based on clinician's judgement, rather than systematic sampling approach. This may result in selection bias. In addition, this laboratory network conducts testing for public health surveillance, including SARI, ILI and SHIVERS-II and WellKiwis cohort surveillance.

**Data analyses.** Study data were captured using REDCap 10.0.19 electronic data capture tools[24]. Analyses were performed in Stata 16.1 (StataCorp LLC).

The observed incidence rates of influenza-PCR-confirmed SARI or ILI or ARI were corrected each week to account for missed swabs from ILI cases by applying the influenza positivity rate of those tested to those non-tested (Corrected number of influenza-PCR-confirmed SARI or ILI or ARI events = Number of SARI or ILI or ARI × Actual number of influenza-PCR-confirmed SARI or ILI or ARI/Actual number of SARI or ILI or ARI swabs).

Based on SARI and ILI surveillance data from 2015 to 2019, the start of the annual influenza season and intensity level of the influenza epidemics was defined by using the Moving Epidemic Method (MEM)[25–27]. Briefly, MEM has three main steps: Step 1: for each season separately, the length of the epidemic period is estimated as the minimum number of consecutive weeks with the maximum accumulated percentage rates, splitting the season into three periods: a pre-epidemic, an epidemic, and a post-epidemic period; Step 2: MEM calculates the epidemic threshold as the upper limit of the 95% one-sided confidence interval of 30 highest pre-epidemic weekly rates, the $n$ highest for each season taking the whole training period, where $n = 30$/number of seasons; Step 3: medium, high, and extra-ordinary intensity thresholds were estimated as the upper limits of the 40, 90, and 97.5% one-sided confidence intervals of the geometric mean of 30 highest epidemic weekly rates, the $n$ highest for each season taking the whole training period, where $n = 30$/number of seasons. Five categories are used to set thresholds and define intensity level as no activity or below epidemic threshold, low (0–40%), moderate (40–90%), high (90–97.5%) and extra-ordinary (>97.5%) one-sided confidence interval of the geometric mean.

Cochran–Armitage test for trend analysis was performed for ICD-coded influenza hospitalisations and numbers of the reported virus detections.

Laboratory-based surveillance data and ICD-coded influenza hospitalisation data used the median weekly value to represent the reference period of 2015–2019. Median and interquartile ranges were calculated for the number of viruses reported during 2015–2019; percentage of reduction = 1 − [No. of virus (2020)/median no. of virus (2015–2019)]%.

**Reporting summary.** Further information on research design is available in the Nature Research Reporting Summary linked to this article.

## Data availability

Anonymised raw data and Stata syntax are used to produce all the analyses, figures and tables for this paper. Source data are provided with this paper. All requests for raw and analysed data will be reviewed by the corresponding authors to verify whether the request is subject to any intellectual property or funder or confidentiality obligations.

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

## Acknowledgements

The SHIVERS-II project is funded by US National Institute of Allergy and Infectious Diseases (NIAID) (CEIRS Contract HHSN272201400006C). The WellKiwis (i.e. SHI-VERS-III) project is funded by US-NIAID (U01 AI 144616). The SARI and ILI surveillance were funded by the NZ Ministry of Health during 2017–2020 and by US Centers for Disease Control and Prevention (U01IP000480) during 2012–2016. The funding resource has no role in study design; collection, analysis or interpretation of data; writing of reports; nor decision to submit papers for publication. SHIVERS-II & Well-Kiwis cohort study, SARI and ILI surveillance, led by the Institute of Environmental Science and Research (ESR), is a multi-centre and multi-disciplinary collaboration. The authors wish to thank SHIVERS collaborating organisations for their commitment and support: ESR, Auckland District Health Board (DHB), Counties Manukau DHB, Capital Coast DHB; Hutt Valley DHB; Regional Public Health; University of Auckland, University of Otago, WHO Collaborating Centre at St Jude Children's Research Hospital in Memphis, USA. Wellington Maternity Health Professionals; sentinel general practices; local influenza coordinators within local Public Health Units; participating hospital virology laboratories in ADHB, CMDHB, Waikato, Wellington, Dunedin and Christchurch. The findings and conclusions in this report are those of the authors and do not necessarily represent the views of the US National Institute of Allergy and Infectious Diseases, US Department of Health and Human Services, the Institute of Environmental Science and Research (ESR) or any other collaborating organisations.

## Author contributions

All authors meet the International Committee of Medical Journal Editors criteria for authorship. Q.S.H., R.W., P.T., B.Y., K.D., A.N., S.J., T.D., N.T., P.C.-S., M.B., H.D., C.C. G., S.J., M.G.B., S.T., A.T., C.W., S.R., C.McArthur., T.K. and N.A. designed and operationalised the SARI, ILI and/or SHIVERS-II&III cohort platforms; L.J., J.R., W.G., J.B., J.D., T.M., S.T., S.R., F.R., J.B., C.Mansell., M.D., A.W., J.G., A.v.d.L. and M.B. provided the testing and reporting; T.J., C.E., W.T., L.E., M.M., A.M., S.J., L.L., B.W., H.S., R.G., K.C., D.C., S.L., A.S. and T.W. did the clinical data and sample collection and reporting and ensured operations; T.W. and Q.S.H. did the data analysis; Q.S.H. wrote the first draft of the manuscript. All authors contributed to the interpretation of the results, revision of the manuscript critically for intellectual content and have given final approval of the version to be published.

## Competing interests

The authors declare no competing interests.

## Additional information

## NPIsImpactOnFlu Consortium

Shirley Lawrence[7] & Alicia Stanley[8]

A full list of members and their affiliations appears in the Supplementary Information.

