## [Peer Review File · Nature Communications]

REVIEWER COMMENTS

Reviewer #1 (Remarks to the Author):

In this paper, Huang et al demonstrate the impact of non-pharmaceutical interventions for COVID-19 on influenza (and other respiratory viruses) in NZ in the winter of 2020. These findings are important and serve as a basis for future recommendations for control of pandemic influenza. This is generally well written manuscript and the analysis are simple, straightforward sound. I have a few comments for the authors:

1. How can you conclusively say that the virus reached NZ on 28 Feb? The first case may have been identified on this but that is not same as when virus reached NZ.
2. This has also been seen in other southern hemisphere countries like Australia, Argentina, Columbia etc. so should be referenced
3. Regarding influenza vaccination, is this overall coverage or coverage in high risk groups. Not clear in this sentence when flu vaccine became part of national immunization policy. Was 2020 the first year or 2019?
4. In the context of influenza A and rhinovirus, should talk about concept of virus interference. Ref Sema Nickbacksh PNAS

Reviewer #2 (Remarks to the Author):

General comment:

This is a very nice paper. Congratulations. It would be nice to publish the paper asap as it is highly relevant to countries in the Northern Hemisphere. I have some suggestions to improve the text.

1. I had a lot of trouble understanding the sentence: 'Influenza virus has a shorter serial interval and earlier peak infectivity compared to SARS-CoV-2. Our recent publication also showed that up to 32% of influenza virus infections in NZ are mild or asymptomatic, suggesting the likelihood of substantial asymptomatic transmission.' I am not sure what the authors mean by 'serial interval' and the issue of substantial asymptomatic transmission is also an issue with COVID-19 (80% have mild infections). Regarding the second point, are they suggesting that flu is less than COVID-19? This is not clear.
2. Discussion. I think the authors need to also talk about data from Australia. There are a couple of interesting references: the Yeoh paper in CID (<https://academic.oup.com/cid/advance-article/doi/10.1093/cid/ciaa1475/5912591>), the Britton et al paper in The Lancet ([https://www.thelancet.com/journals/lanchi/article/PIIS2352-4642\(20\)30307-2/fulltext](https://www.thelancet.com/journals/lanchi/article/PIIS2352-4642(20)30307-2/fulltext)) and the NSW surveillance reports (<https://www.health.nsw.gov.au/Infectious/covid-19/Pages/weekly-reports.aspx>)
3. Discussion: The Britton paper (Australia) provides a theory for why Rhinoviruses are less affected by the COVID-19 control measures: 'Rhinoviruses are easily transmitted between children in close contact and are non-enveloped so might be inherently less susceptible to inactivation by handwashing.' I think the authors should consider this point in their paper.
4. Discussion. The NSW surveillance reports indicate that RSV is increasing strongly at the end of the winter. It would be very interesting for the NZ group to comment about this. It doesn't look like NZ is seeing a similar pattern. Or is this because schools have been opened in NSW and not in NZ?
5. Discussion: The authors make the following statement: 'We postulate that NZ's use of stringent NPIs (lockdowns and border controls) have markedly changed human behaviour, resulting in substantial reductions in contacts between influenza-infected individuals and influenza-susceptible individuals.' I find it surprising that the research group can collate such detailed data on respiratory infections but not provide (some) behavioural data to support this statement. There must be some behavioural data for NZ to support this (e.g. the Google movement data).
6. Discussion. I miss a section about the role of children in the Discussion. If I understand rightly schools have been completely closed in NZ (full lockdown period). Has this continued? If not, have

they seen an increase in some respiratory infections (e.g. RSV, like in NSW)?

7. Discussion: Congratulations with the paragraph about WHO's pandemic influenza intervention guidance. This is well written and super relevant. I was very happy to see this paragraph.

8. Discussion: One important issue regarding influenza is whether laboratories are totally focused on COVID-19 and they are not testing for influenza (and other respiratory viruses). A statement is made about this point in the study limitations: 'Secondly, during the COVID-19 laboratory response, some laboratories prioritised testing for COVID-19 and reduced testing for influenza and other respiratory viruses.' Considering this is a surveillance group with access to detailed data, I think this hypothesis could be tested. Why don't they compare the tested specimen numbers over time? Has the number of specimens tested for influenza changed massively compared to previous years?

Rebuttal to REVIEWER COMMENTS

Reviewer #1 (Remarks to the Author):

In this paper, Huang et al demonstrate the impact of non-pharmaceutical interventions for COVID-19 on influenza (and other respiratory viruses) in NZ in the winter of 2020. These findings are important and serve as a basis for future recommendations for control of pandemic influenza.

This is generally well written manuscript and the analysis are simple, straightforward sound. I have a few comments for the authors:

1. How can you conclusively say that the virus reached NZ on 28 Feb? The first case may have been identified on this but that is not same as when virus reached NZ.

Rebuttal: We would like to thank the reviewer for the comment. The reviewer is correct. We cannot know for sure the date that the virus first reached New Zealand. We changed lines 50&51 to: “The coronavirus disease 2019 (COVID-19), declared a pandemic by the World Health Organization (WHO) on 11 March 2020, was first identified in a person in New Zealand (NZ) on 28 February 2020.”

2. This has also been seen in other southern hemisphere countries like Australia, Argentina, Columbia etc. so should be referenced

Rebuttal: We would like to thank the reviewer for the comment and added reference for other southern hemisphere countries. We changed lines 156&157:

“NZ data, presented here, is consistent with what reported from other southern hemisphere countries^{1,2} in Australia, Chile and South Africa, as well as reported from Hong Kong during the 2003 SARS epidemic,³ and the COVID-19 pandemic.⁴ Therefore, we suggest it is important to re-evaluate the role of stringent NPIs such as lockdowns and border closures in mitigating or even eliminating severe pandemic influenza. Although such measures are associated with significant negative impacts on society, their potential beneficial effects on delaying, containing or averting transmission and saving lives should be assessed. New knowledge from this assessment may inform better preparedness for future influenza pandemics and other severe respiratory viral threats. Additionally, it would be a worthwhile endeavour to conduct detailed analysis to identify which components of NPIs were most effective for preventing seasonal influenza and other respiratory virus infection and transmission. Careful investigation of NPIs may identify new and sustainable interventions that can minimize and prevent seasonal and epidemic respiratory viral illnesses in the future.”

3. Regarding influenza vaccination, is this overall coverage or coverage in high risk groups. Not clear in this sentence when flu vaccine became part of national immunization policy. Was 2020 the first year or 2019?

Rebuttal: flu vaccine became part of national immunization policy in New Zealand since 1997.

We amended lines 164&165: “The NZ National Immunisation Register recorded ~22% influenza vaccine coverage in 2020 (35% more influenza vaccinations recorded during April-June in 2020 compared with 2019, personal communication)

4. In the context of influenza A and rhinovirus, should talk about concept of virus interference. Ref Sema Nickbakhsh PNAS

Rebuttal: We thank the reviewer for the comment and have now cited the reference on viral interference by Nickbakhsh et al. We have amended lines 170&174:

“Rhinovirus, the most prevalent virus after the lockdown, may provide viral interference and thus reduce the risk of individuals being infected with influenza viruses. Interferon-stimulating immunity mediated by rhinovirus infection may make it difficult for additional viruses such as influenza to become established in a population.⁵ Similar observations were reported in Sweden and France during the 2009 H1N1 pandemic that the annual autumn rhinovirus epidemic interrupted and delayed community transmission of the emerging influenza virus.^{6,7}”

Reviewer #2 (Remarks to the Author):

General comment:

This is a very nice paper. Congratulations. It would be nice to publish the paper asap as it is highly relevant to countries in the Northern Hemisphere. I have some suggestions to improve the text.

1. I had a lot of trouble understanding the sentence: ‘Influenza virus has a shorter serial interval and earlier peak infectivity compared to SARS-CoV-2. Our recent publication also showed that up to 32% of influenza virus infections in NZ are mild or asymptomatic, suggesting the likelihood of substantial asymptomatic transmission.’⁶ I am not sure what the authors mean by ‘serial interval’ and the issue of substantial asymptomatic transmission is also an issue with COVID-19 (80% have mild infections). Regarding the second point, are they suggesting that flu is less than COVID-19? This is not clear.

Rebuttal: We would like to thank the reviewer for the comment. We amended lines 70&73:

“Influenza virus has a short serial interval (the mean interval between illness onset in two successive patients in a chain of transmission) of 2-4 days. Viral excretion peaks early in the illness (i.e. during the first 1-3 days of illness). These features of influenza infection mean there is limited time to effectively implement isolation and quarantine measures. Additionally, substantial asymptomatic infection⁸ creates difficulties in finding cases to initiate nonpharmaceutical measures. These characteristics have led to the assumption that these NPIs would not be effective in controlling influenza virus⁹.”

2. Discussion. I think the authors need to also talk about data from Australia. There are a couple of interesting references: the Yeoh paper in CID

(<https://academic.oup.com/cid/advance-article/doi/10.1093/cid/ciaa1475/5912591>), the

Britton et al paper in The Lancet

([https://www.thelancet.com/journals/lanchi/article/PIIS2352-4642\(20\)30307-2/fulltext](https://www.thelancet.com/journals/lanchi/article/PIIS2352-4642(20)30307-2/fulltext)) and

the NSW surveillance reports (<https://www.health.nsw.gov.au/Infectious/covid-19/Pages/weekly-reports.aspx>)

Rebuttal: We would like to thank the reviewer for the comment and reference to the recent publication by Britton¹⁰ et al that have reported the impact of COVID-19 public health measures on presentations to the Sydney Children's Hospitals Network with respiratory syncytial virus infections. We also cited the reference by Yeoh² et al that have reported respiratory syncytial virus and influenza detections in Western Australian children.

We have now cited both of these references in our reference to data reported from other southern hemisphere countries (see our response above to Reviewer 1, Question 2) and the role of handwashing in prevention of spread of non-enveloped viruses (see our response to Reviewer 2, Question 3).

3. Discussion: The Britton paper (Australia) provides a theory for why Rhinoviruses are less affected by the COVID-19 control measures: 'Rhinoviruses are easily transmitted between children in close contact and are non-enveloped so might be inherently less susceptible to inactivation by handwashing.' I think the authors should consider this point in their paper.

Rebuttal: We would like to thank the reviewer for directing us to this important recent reference. Hand washing results in the removal of dirt, organic material and transient microorganisms. During hand washing, friction is created and along with soap and water, this action removes soiling. Alcohol-based hand rubs have activity against non-enveloped viruses such as rhinovirus. Rhinovirus may be less susceptible to inactivation by soap-and-water type of hand washing. Additionally, children's generally poor quality of handwashing may also be another contributing factor.

We amended lines 178&180:

"Rhinovirus infections, responsible for more than one-half of cold-like illnesses, are frequently transmitted within households from children to other family members.¹¹ Additionally, rhinoviruses are non-enveloped viruses so might be inherently less susceptible to inactivation by soap-and-water handwashing.¹⁰ Furthermore, the quality of children's handwashing is likely to be poor. These factors may have contributed to rhinovirus infection being less affected by the COVID-19 control measures."

4. Discussion. The NSW surveillance reports indicate that RSV is increasing strongly at the end of the winter. It would be very interesting for the NZ group to comment about this. It doesn't look like NZ is seeing a similar pattern. Or is this because schools have been opened in NSW and not in NZ?

Rebuttal: The reviewer is correct that we did not see RSV increase during the 2020 winter season in NZ. Schools have been fully open throughout NZ since 13 May 2020 (note: Auckland had a regional lockdown at Alert level 3 in August 2020 and schools may not function as normal as other regions.) It appears that school opening in late winter months in NZ was not associated with the increase of RSV activity at the end of the winter.

5. Discussion: The authors make the following statement: ‘We postulate that NZ’s use of stringent NPIs (lockdowns and border controls) have markedly changed human behaviour, resulting in substantial reductions in contacts between influenza-infected individuals and influenza-susceptible individuals.’ I find it surprising that the research group can collate such detailed data on respiratory infections but not provide (some) behavioural data to support this statement. There must be some behavioural data for NZ to support this (e.g. the Google movement data).

Rebuttal: The evidence of behavioural change during NZ’s lockdown has been documented in the supplementary figure 2 in the publication¹² (Jefferies, S., *et al.* COVID-19 in New Zealand and the impact of the national response: a descriptive epidemiological study. *Lancet Public Health* Jefferies, S., *et al.* COVID-19 in New Zealand and the impact of the national response: a descriptive epidemiological study. *Lancet Public Health* **5**, e612-e623 (2020))

We have cited this reference in this sentence in line 145: “We postulate that NZ’s use of stringent NPIs (lockdowns and border controls) have markedly changed human behaviour¹², resulting in substantial reductions in contacts between influenza-infected individuals and influenza-susceptible individuals.”

6. Discussion. I miss a section about the role of children in the Discussion. If I understand rightly schools have been completely closed in NZ (full lockdown period). Has this continued? If not, have they seen an increase in some respiratory infections (e.g. RSV, like in NSW)?

Rebuttal: The situation in NZ’s Schools has differed during different alert levels:

- During NZ’s full nationwide lockdown period (Alert level 4) 25-March to 27-April 2020, all educational facilities were closed.
- During level 3 (28-Apr to 12-May), Schools were not fully open. The government guideline is: Schools (years 1 to 10) and Early Childhood Education centres can safely open, but will have limited capacity. Children should learn at home if possible.
- Since 13 May at alert levels 1&2, schools have been mostly open throughout NZ except in Auckland during their local lockdown in August 2020.

Unlike NSW where an increase in RSV detections occurred at the tail end of the winter season, NZ did not see any increase of RSV detections during the whole 2020 winter season.

7. Discussion: Congratulations with the paragraph about WHO’s pandemic influenza intervention guidance. This is well written and super relevant. I was very happy to see this paragraph.

Rebuttal: Many thanks for this comment☺

8. Discussion: One important issue regarding influenza is whether laboratories are totally focused on COVID-19 and they are not testing for influenza (and other respiratory viruses). A statement is made about this point in the study limitations: ‘Secondly, during the COVID-19 laboratory response, some laboratories prioritised testing for COVID-19 and reduced testing for influenza and other respiratory viruses.’ Considering this is a surveillance group with access to detailed data, I think this hypothesis could be tested. Why don’t they compare the tested specimen numbers over time? Has the number of specimens tested for influenza

changed massively compared to previous years?

Rebuttal: The laboratory-based surveillance consists of two components: the first being samples collected for surveillance of severe acute respiratory infections (SARI) and influenza like illnesses in adult and infants in the community (SHIVERS-II&III surveillance, <https://www.esr.cri.nz/our-research/research-projects/shivers-project/>); and the second being samples ordered by clinicians for patient management purposes.

We have detailed specimen numbers over time for the first component. For example, SHIVERS-II&III surveillance doubled its sample testing number because we expanded the case definition from strict influenza-like illness (cough and fever) to any acute respiratory illness (ARI). Regarding SARI surveillance, we maintained the same case definition¹³ during 2015-2020; “an acute respiratory illness with a history of fever or measured fever of $\geq 38^{\circ}\text{C}$, and cough, and onset within the past 10 days, and requiring inpatient hospitalization”. During 2020 only 428 samples from patients with SARI have been collected for testing, compared to median annual number of 1689 tested samples during 2015-2019. In New Zealand there were fewer hospitalisations due to respiratory illnesses in 2020, compared with the numbers hospitalized each year from 2015-2019. This resulted in fewer hospitalized patients meeting the SARI case definition required for sampling. During 2020 we did not have any influenza virus detections among patients with SARI.

However, for those samples ordered by clinicians for patient management purposes, we do not have complete information on the number of all specimens tested. Thus we could not do the comparison as suggested by the reviewer.

1. Olsen, S.J., *et al.* Decreased Influenza Activity During the COVID-19 Pandemic - United States, Australia, Chile, and South Africa, 2020. *Mmwr* **69**, 1305-1309 (2020).
2. Yeoh, D.K., *et al.* The impact of COVID-19 public health measures on detections of influenza and respiratory syncytial virus in children during the 2020 Australian winter. *Clinical Infectious Diseases* <https://doi.org/10.1093/cid/ciaa1475> (2020).
3. Lo, J.Y., *et al.* Respiratory infections during SARS outbreak, Hong Kong, 2003. *Emerg Infect Dis* **11**, 1738-1741 (2005).
4. Cowling, B.J., *et al.* Impact assessment of non-pharmaceutical interventions against coronavirus disease 2019 and influenza in Hong Kong: an observational study. *Lancet Public Health* **5**, e279-e288 (2020).
5. Nickbakhsh, S., *et al.* Virus-virus interactions impact the population dynamics of influenza and the common cold. *Proc Natl Acad Sci U S A* (2019).
6. Casalegno, J.S., *et al.* Impact of the 2009 influenza A(H1N1) pandemic wave on the pattern of hibernal respiratory virus epidemics, France, 2009. *Euro Surveill* **15**(2010).
7. Linde, A., Rotzen-Ostlund, M., Zwegberg-Wirgart, B., Rubinova, S. & Brytting, M. Does viral interference affect spread of influenza? *Euro Surveill* **14**(2009).
8. Huang, Q.S., *et al.* Risk Factors and Attack Rates of Seasonal Influenza Infection: Results of the Southern Hemisphere Influenza and Vaccine Effectiveness Research and Surveillance (SHIVERS) Seroepidemiologic Cohort Study. *J Infect Dis* **219**, 347-357 (2019).
9. World_Health_Organization_Writing_Group, *et al.* Non-pharmaceutical interventions for pandemic influenza, international measures. *Emerg Infect Dis* **12**, 81-87 (2006).
10. Britton, P.N., *et al.* COVID-19 public health measures and respiratory syncytial virus. *Lancet Child Adolesc Health* **4**, e42-e43 (2020).

11. Peltola, V., *et al.* Rhinovirus transmission within families with children: incidence of symptomatic and asymptomatic infections. *J Infect Dis* **197**, 382-389 (2008).
12. Jefferies, S., *et al.* COVID-19 in New Zealand and the impact of the national response: a descriptive epidemiological study. *Lancet Public Health* **5**, e612-e623 (2020).
13. Huang, Q.S., *et al.* Southern Hemisphere Influenza and Vaccine Effectiveness Research and Surveillance. *Influenza Other Respir Viruses* **9**, 179-190 (2015).